# Training Data Eigenvector Dynamics in the EigenPro Implementation of the Neural Tangent Kernel and Recursive Feature Machines

**Cyril Gorlla**
Halıcıoğlu Data Science Institute
University of California San Diego
La Jolla, CA 92093, USA
`cyril.m.gorlla@jacobs.ucsd.edu`

## Abstract

There has been much recent work on kernel methods as a viable alternative to deep neural networks (DNNs). The advent of the *Neural Tangent Kernel* (NTK) has brought on renewed interest in these methods and their application to typical deep learning tasks. Recently, kernels have been shown to be capable of feature learning similar to that of DNNs, termed *Recursive Feature Machines* (RFMs). In accordance with the growing scale of kernel models, the EigenPro 3 algorithm was proposed to facilitate large-scale training based on preconditioned gradient descent. We propose an accessible framework for observing the eigenvector dynamics of EigenPro's training data in its implementation of these kernel methods, and find empirically that significant change ceases early in training along with apparent bias towards equilibrium. In the case of RFMs, we find that significant change in the training data eigenvectors typically curtails before five iterations, in accordance with findings that RFMs achieve optimal performance in five iterations. This represents a path forward in gaining intuition for the inner workings of large-scale kernel training methods. We provide an easy to use Python implementation of our framework at `https://github.com/cgorlla/ep3dynamics`.

## 1 Introduction

In the past decade, deep neural networks have achieved state of the art performance in a wide variety of areas, including computer vision (Szegedy et al., 2013), speech synthesis (Ze et al., 2013), and text generation (Brown et al., 2020). More recently, it has been observed that increasing sample and parameter count in deep learning models yields significant gains in model performance (Chowdhery et al., 2022), bucking classical statistical theory which suggests that overfitting should cause poorer performance (Belkin et al., 2019). Considering this and other phenomena, it is apparent that the advances in deep learning are rapidly outpacing our understanding of *why* they perform so well. Kernels have been a well studied concept for over a century (Aronszajn, 1950), and have recently found relevance in machine learning spurred by the *Neural Tangent Kernel* (Jacot et al., 2018), which showed that the behavior of a fully-connected neural network can be captured by the NTK. Consequently, it has been suggested that, as kernels display similar phenomena to over-parameterized neural networks, and since kernels are easier to analyse than neural networks (Belkin et al., 2018), understanding kernel methods will yield insights for deep learning as well.

To that end, there has been much recent work analyzing kernel methods in relation to DNNs. The evolution of the NTK has been shown to consist of a rapid initial transient in the first few epochs (Fort et al., 2020), and has the propensity to stay constant in deep networks when in an "ordered" hyperparameter phase (Poole et al., 2016; Seleznova & Kutyniok, 2022). Furthermore, feature learning in fully connected neural networks was shown to be connected to a statistical object called the expected gradient outer product, and this property was exploited in a kernel framework termed *Recursive Feature Machines* (Radhakrishnan et al., 2022). RFMs were shown to surpass state of the art performance on a plethora of tabular dataset benchmarks. EigenPro 3 (Abedsoltan et al., 2023) is a recently proposed algorithm to facilitate the efficient training of large-scale kernel models utiliz-

ing projected dual preconditioned stochastic gradient descent. We use EigenPro 3 for the NTK and EigenPro 2 for the RFM, as this is the currently available implementation. Considering the growing promise of kernel methods, building accessible frameworks to understand large-scale kernel training algorithms is prescient. Consequently, we explore the dynamics of eigenvectors in training data in EigenPro.

## 2 SETUP

For a general machine learning problem, we have training data $(x, y) = x_i \in \mathbb{R}^d, y_i \in \mathbb{R}^n_{i=1}$. This training data is used by a machine learning model to generate some predicted output. A general kernel model predicts with $f(x) = \sum_{i=1}^p \alpha_i K(x, z_i)$, where $K$ is a positive semi-definite kernel, and $z_i$ are (arbitrary) model centers. EigenPro 3 (and 2) utilizes Nyström approximation in order to compute inexact projection, based on the Nyström extension (Ma & Belkin, 2019): $\psi_i = K(\cdot, X) \frac{e_i}{\sqrt{\lambda_i}} = \sum_{j=1}^n K(\cdot, x_j) \frac{e_i}{\sqrt{\lambda_i}}$, where $\lambda_i$ is an eigenvalue of the Hessian operator for the square loss for $1 \le i \le n$ and $\psi_i$ its eigenfunction, such that Hessian Operator $= \lambda_i \psi_i$. Following this, $\lambda_i$ is also an eigenvalue of $K(x, x)$ and consequently if $e_i$ is a unit-norm eigenvector, $K(x, x) \cdot e_i = \lambda_i e_i$, leading to the above equation. It is in this step of EigenPro that we extract the eigenvectors with respect to the training data at a given epoch. As RFMs operate on iterations with multiple epochs, we modify the Nyström approximation and iterative solver in the EigenPro implementation of RFMs to expose the training data eigenvectors at the end of each iteration. The implementation of the NTK is with respect to fully-connected neural networks with ReLU nonlinearity.

## 3 RESULTS

CIFAR-10 was used to train the models. For the NTK, the number of model centers was 5000 and a depth of 2 was used, resulting in 5000 eigenvectors. These values were chosen for the purposes of these experiments, but similar phenomena were observed with larger models with 10000 and 15000 centers. 2000 samples were selected by EigenPro 3 for projection in each epoch. For the RFM, 5000 samples were used and, similarly, 2000 subsamples were used at each iteration of EigenPro 2, resulting in 2000 eigenvectors. For an eigenvector e, we compute the change in e from time $i$ to $j$ as follows: $\sum^n e_j - \sum^n e_i$. We plot the eigenvectors at each epoch/iteration with respect to the change from the original eigenvector. We observe both attractive and repulsive effects in the distribution of training data evolution for both NTK and RFM models. If e.g. positive or negative eigenvectors are more spread out at one point in time, this effect is reversed in the next, and vice versa. For the NTK, we see that overall changes in the eigenvectors are minute after the first epoch, which is in line with the nature of the initialization of the EigenPro 3 algorithm (Abedsoltan et al., 2023). This holds true for the RFM, albeit with more variability. Unlike the NTK, whose capability to learn features has been questioned (Yang & Hu, 2020), the RFM kernelizes a mechanism provably linked to feature learning in fully-connected neural networks. Empirically, RFMs were shown to attain optimal performance in many scenarios within five iterations (Radhakrishnan et al., 2022); in the EigenPro implementation of RFMs, we also did not typically observe significant change in training data eigenvectors after five iterations.

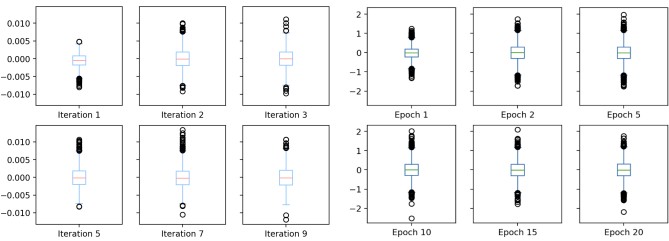

Figure 1: Boxplots of eigenvectors (Left: RFM, Right: NTK)

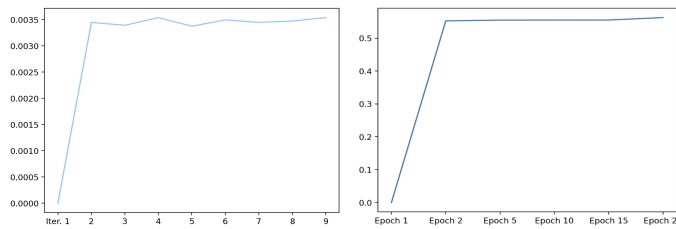

Figure 2: Change from original eigenvectors (Left: RFM, Right: NTK)

URM STATEMENT

This paper meets the criteria of the ICLR 2023 Tiny Papers Track.

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

## A  APPENDIX

### A.1  DEEPER NTK

When using a depth of 20 for the NTK, we do not observe any major changes in the phenomena observed.

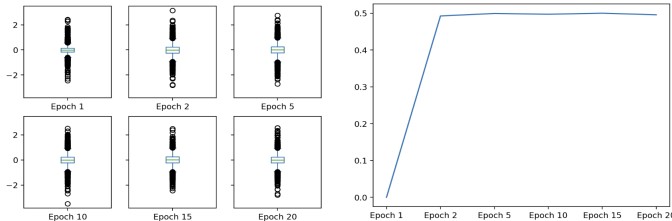

Figure 3: Left: Boxplot of eigenvectors, Right: Change in eigenvectors

### A.2  CODE

To illustrate the intuitive nature of our framework, we provide example code. Code can be found on the GitHub repository: `https://github.com/cgorlla/ep3dynamics`.

```python
import torch
from eigenpro3.utils import accuracy, load_dataset
from eigenpro3.datasets import CustomDataset
from eigenpro3.models import KernelModel
from eigenpro3.kernels import ntk_relu
import os

from torchvision.datasets import CIFAR10
os.environ['DATA_DIR'] = './download'
CIFAR10(os.environ['DATA_DIR'], train=True, download=True)

p = 5000 # model size

if torch.cuda.is_available():
    DEVICES = [torch.device(f'cuda:{i}') for i in range(torch.cuda.
    device_count())]
else:
    DEVICES = [torch.device('cpu')]

evs = []

for num in [epochs]:
```

```
23
24      kernel_fn = lambda x, z: ntk_relu(x, z, depth=2)
25
26      n_classes, (X_train, y_train), (X_test, y_test) = load_dataset('
        cifar10')
27
28      centers = X_train[torch.randperm(X_train.shape[0])[:p]]
29
30      testloader = torch.utils.data.DataLoader(
31          CustomDataset(X_test, y_test.argmax(-1)), batch_size=512,
32          shuffle=False, pin_memory=True)
33
34      model = KernelModel(n_classes, centers, kernel_fn, X=X_train, y=
        y_train, devices=DEVICES)
35      model.fit(model.train_loaders, testloader, score_fn=accuracy, epochs=
        num)
36      evs.append(model.eigenvectors_data)
```

