# OpenReview forum: "Training Data Eigenvector Dynamics in the EigenPro Implementation of the Neural Tangent Kernel and Recursive Feature Machines"
_ICLR.cc/2023/TinyPapers — Submitted to Tiny Papers @ ICLR 2023_

### Official Review · Reviewer_hY7R · 2023-03-26

**Confidence:** 3

**Summary Of Contributions:**

This study evaluates the dynamic change of eigenvectors in NTK and RFM and finds RFM achieves optimal performance in 5 iterations.

**Rating:**

Clear, Correct, and Reproducible (CCR): a submission which meets the reviewing criteria

**Strengths And Weaknesses:**

Strengths

1. The paper is clearly written with comprehensive literature review and math details.

2. The experiment design is valid.

3. Python code is provided to replicate the result.

Weaknesses

1. The authors described the observation clearly but didn't explain why it happen. For example, what are the potential reasons for the "repulsive and attractive effects"?

2. The authors mentioned "in accordance with findings that RFMs achieve optimal performance in five iterations", what are those findings? Please provide references. Why is 5 the magic number? What are potential reasons?

**Suggested Changes:**

Suggestions

1. For visualization, I would suggest using two different colors for RFM and NTK results separately.

2. If you have room, I would suggest highlighting specific broader impact of this work.

3. A potential minor mistake: I think the notation should be $y_i \in \mathbb{R}^n$ in the first sentence of Section 2.

4. Please kindly explain why 5000 was used as model centers and 2000 was used as subsamples.

5. Please explain the math formula of how change is measured in figures.

6. It may be worth plotting change after 5 epochs when claiming "we did not observe significant change in training data eigenvectors after five iterations."

---

> ### Author Response · Authors · 2023-05-05
> **Response to ICLR 2023 TinyPapers Paper235 Reviewer hY7R**
>
> Hello, thank you for your valuable comments.
>
> We have clarified the observations and the nature of previous work on RFMs reaching optimal performance. We have also differentiated the colors as suggested, and explained the choice of model size. We clarified how change is measured, and expanded the plot of the NTK to better show change after 5 iterations. Consequently, we believe we have addressed your concerns.
>
> With regards to suggestion #3, the $x_i$s need not necessarily be in the same dimension as the $y_i$s.

---

### Official Review · Reviewer_J2wD · 2023-03-30

**Confidence:** 3

**Summary Of Contributions:**

The paper proposes a framework for observing the eigenvector dynamics of EigenPro 3's training data in its implementation of kernel methods. The framework shows that significant change in the training data eigenvectors typically curtails early in training, suggesting bias towards equilibrium.

**Rating:**

Clear, Correct, and Reproducible (CCR): a submission which meets the reviewing criteria

**Strengths And Weaknesses:**

## Strengths

- Provides a clear introduction to the motivation for understanding kernel methods and their application to deep learning.
- Explains the concept of EigenPro 3 and how it facilitates large-scale training based on preconditioned gradient descent.
- Discusses the relevance of the Neural Tangent Kernel and how it relates to the behavior of fully-connected neural networks.
- Provides an accessible framework for observing the eigenvector dynamics of EigenPro 3's training data and finding empirical evidence of change in the training data eigenvectors.
- Provides an easy-to-use Python implementation of the framework.
- Discusses the significance of the findings for gaining intuition into the inner workings of large-scale kernel training methods.

## Weaknesses

- The submission could benefit from more specific details on the results and their implications for the field.
- The submission could provide more context on the significance of the findings for the development of kernel-based machine learning algorithms.
- While the proposed framework for observing eigenvector dynamics in EigenPro 3 and RFMs is new and valuable, it does not necessarily introduce a fundamentally new idea to the field. However, it still contributes to the growing body of knowledge on kernel methods and their application to deep learning, and provides a useful tool for understanding the inner workings of large-scale kernel training algorithms.
- Not anonymized properly, the code implementation is available in a public GitHub repository.

**Suggested Changes:**

Overall, the submission is well-written and provides a clear overview of recent work on kernel methods in machine learning, focusing specifically on the EigenPro 3 algorithm and its use in Recursive Feature Machines (RFMs). However, there are a few suggestions that the author(s) could consider to improve the submission:

- While the submission provides a clear overview of recent work on kernel methods, it is not immediately clear what the authors' contributions are. It would be helpful to explicitly state the contributions of the paper in the introduction or abstract.
- The submission assumes some familiarity with the Neural Tangent Kernel (NTK) and Recursive Feature Machines (RFMs). It would be helpful to provide more context and background information for readers who may not be as familiar with these concepts.
- The submission briefly mentions that the authors "explore the dynamics of eigenvectors in training data in EigenPro 3" but does not provide much detail on what this entails or how it relates to the larger discussion on kernel methods in machine learning. Providing more detail on this would be helpful for readers.
- The submission uses some vague or imprecise language at times (e.g., "significant change," "apparent bias towards equilibrium"). Using more specific language would make the submission clearer and more precise.
- Consider including more concrete results: While the submission mentions that the authors found that "significant change in the training data eigenvectors typically curtails before five iterations" in the case of RFMs, it does not provide much detail on this or any other results. Including more concrete results would make the submission more compelling and informative.
- The submission mentions that the authors provide an "easy to use Python implementation" of their framework, but does not provide more details on the implementation description. Providing more detail on the implementation and/or a link to the code repository would be helpful for readers.

---

> ### Author Response · Authors · 2023-05-05
> **Response to ICLR 2023 TinyPapers Paper235 Reviewer J2wD**
>
> Hello, thank you for your comments.
>
> We've provided more details on the observed phenomena, and clarified language throughout with regards to our contributions. We also clarified the relation with previous work and provided more details on the experimental setup. We also provided a link to the repository with the code. With these changes, we believe we've substantively addressed your concerns.
>
> With regards to assuming familiarity with the subject matter, we believe that the references given are sufficient for this in light of the formatting constraints.

---

### Comment · Area_Chair_d9bK · 2023-06-06
**Check for Archival**

This work meets the threshold for archival, contents the URM statement and is deanonymized.

---

### Meta-Review · Area_Chair_d9bK · 2023-04-08

**Recommendation:** Invite to present
**Confidence:** 4

**Metareview:**

This paper tries to empirically analyze the training behaviors of large-scale kernel training methods. Specifically, the authors propose a framework for observing the eigenvector dynamics of EigenPro 3's training data in its implementation of kernel methods. The authors empirically observe significant change ceases early in training along with apparent bias towards equilibrium. In RFMs, the rapid change typically ceases before five iterations, where RFMs achieve optimal performance.

Both reviewers find that this paper is well-motivated and clearly written. The studied topic is interesting and timely. The authors further provide the Python script. Therefore, it clearly meets the review criterion.

The reviewers also give many constructive suggestions to improve the paper further. Please carefully revise and proofread the paper following the comments. The AC believes it would be a good submission after revision. Some major points are summarized as follows:
* describe the contribution explicitly in Introduction; add preliminaries in the Appendix;

* add more details and explanations (see reviewers' comments for more details);

* insights about the empirical observations, and the potential impact/implications for the development of kernel-based machine learning algorithms.

Overall, based on the review criteria of the ICLR TinyPaper Track, it meets the CCR standard. We recommend the acceptance of this paper.


**Summary:**

This paper tries to empirically analyze the training behaviors of large-scale kernel training methods. Specifically, the authors propose a framework for observing the eigenvector dynamics of EigenPro 3's training data in its implementation of kernel methods. The authors empirically observe significant change ceases early in training along with apparent bias towards equilibrium. In RFMs, the rapid change typically ceases before five iterations, where RFMs achieve optimal performance.

**Comments And Feedback To The Authors:**

Please carefully revise and proofread the paper following both reviewers' comments.

**Reason For Not Giving A Higher Recommendation:**

* Some concerns exist, which could significantly improve the paper. Please see the reviewers' comments.

**Reason For Not Giving A Lower Recommendation:**

* This paper is well-motivated and clearly written.

* Good Reproducibility: the Python script is provided.

---

> ### Author Response · Authors · 2023-05-05
> **Response to ICLR 2023 TinyPapers Paper235 Area Chair d9bK**
>
> We appreciate all of the reviewers' comments. We have ameliorated many of the issues stated and believe the concerns presented no longer pose an issue.

---

### Decision · Program_Chairs · 2023-04-09

Invite to present